# An Accuracy-Aware Energy-Efficient Multipath Routing Algorithm for WSNs

**DOI:** 10.3390/s24010285

**Published:** 2024-01-03

**Authors:** Feng Dan, Yajie Ma, Wenqi Yin, Xian Yang, Fengxing Zhou, Shaowu Lu, Bowen Ning

**Affiliations:** 1Engineering Research Center for Metallurgical Automation and Measurement Technology of Ministry of Education, Wuhan University of Science and Technology, Wuhan 430081, China; danfeng@wust.edu.cn (F.D.); yinwq980825@163.com (W.Y.); zhoufengxing@wust.edu.cn (F.Z.); shawn2013@wust.edu.cn (S.L.); ningbwen@wust.edu.cn (B.N.); 2Alliance Business School, University of Manchester, Manchester M13 9PL, UK; xian.yang@manchester.ac.uk

**Keywords:** artificial immune algorithm, data accuracy, energy-efficient, multipath routing, wireless sensor networks

## Abstract

In the fields of industrial production or safety monitoring, wireless sensor networks are often content with unreliable and time-varying channels that are susceptible to interference. Consequently, ensuring both transmission reliability and data accuracy has garnered substantial attention in recent years. Although multipath routing-based schemes can provide transmission reliability for wireless sensor networks, achieving high data accuracy simultaneously remains challenging. To address this issue, an Energy-efficient Multipath Routing algorithm balancing data Accuracy and transmission Reliability (EMRAR) is proposed to balance the reliability and accuracy of data transmission. The multipath routing problem is formulated into a multi-objective programming problem aimed at optimizing both reliability and power consumption while adhering to data accuracy constraints. To obtain the solution of the multi-objective programming, an adaptive artificial immune algorithm is employed, in which the antibody initialization method, antibody incentive calculation method, and immune operation are improved, especially for the multipath routing scheme. Simulation results show that the EMRAR algorithm effectively balances data accuracy and transmission reliability while also saving energy when compared to existing algorithms.

## 1. Introduction

Wireless sensor networks (WSNs) are composed of many small, energy-limited, and low-cost sensor nodes deployed in a certain area, forming a multihop self-organizing network system through wireless communication. In fields like industrial production or safety monitoring, WSN applications have high requirements for data quality. In these applications, the reliability of data transmission and the accuracy of received data directly affect the response and handling of user applications to emergencies such as fire or toxic gas leakage. However, in environments like factories or outdoors, the wireless channels used in WSN communication exhibit the characteristics of unreliability, time variation, and susceptibility to interference [1,2]. These characteristics pose challenges in ensuring the reliability of the transmission path and the accuracy of received data, leading to widespread concerns in recent years [3].

An important technology for ensuring transmission reliability lies in the deployment of highly reliable routing algorithms [4,5,6], with multipath routing standing out as a widely adopted strategy to improve the reliability of data transmission in WSNs. In contrast to single-path routing, multipath routing augments data transmission reliability by establishing multiple paths between source and destination nodes. To achieve greater throughput, enhanced failure resilience, faster mobility and reduced latency, coding-based multipath routing schemes have garnered increasing research attention in recent years [7,8,9]. These routing schemes typically segment the original data into multiple coding packets in uniform size, containing some redundant information. The coding packets are then distributed across multiple paths for transmission. Intermediate nodes process the data through techniques such as data aggregation or compression to reduce the number of packets. Once a certain quantity of coding packets is collected, the destination node recovers the original data through decoding and data fusion [10]. However, it is worth noting that the aggregation of multiple coding packets at intermediate or sink nodes along the path may lead to a reduction in data accuracy [11,12]. This reduction in accuracy can have repercussions on the result of data analysis or decision-making. Consequently, maintaining data accuracy poses a challenge for coding-based multipath routing.

To address this challenge, current research efforts can be categorized into two main approaches. One approach focuses on improving the routing schemes to reduce the number of data copies by reasonably selecting or minimizing the transmission paths. In the literature [13], a combination of multipath routing and a tree-based aggregation algorithm was employed. During path construction, a node is selected and remains as the first node of aggregation, and convergence proceeds step by step from this node to the destination. In the literature [14], the authors combined the clustering method with multipath routing and utilized spatio-temporal correlation to aggregate sampled data at the source node to reduce the volume of transmitted data. The second approach to improving aggregation algorithms is to guarantee data accuracy by reducing the accuracy loss. Fan and Chen proposed a Scalable Counting (SC) algorithm to guarantee data accuracy [11]. The SC algorithm inherits the idea of the Linear Counting (LC) algorithm, which incorporates a hash function with arrays to reduce the accuracy loss caused by aggregation calculation. In the literature [15], a routing tree is established for data transmission. It calculates the maximum allowable data accuracy loss of each subtree in proportion to the size of the routing tree. The data compression ratio at each node is then determined based on the maximum allowable accuracy loss to ensure accuracy. However, there is a trade-off between data accuracy and data transmission reliability [11]—more coding packets can enhance the reliability of data transmission, whereas multiple aggregations of copies during routing can reduce data accuracy at the destination. As aggregation instances increase, the loss of data accuracy becomes more pronounced, which makes the accuracy-aware data collection algorithms less effective.

To tackle the aforementioned challenge, a multipath routing algorithm EMRAR (Energy-efficient Multipath Routing algorithm balancing data Accuracy and transmission Reliability) is proposed in this paper. The core concept behind EMRAR is to jointly optimize transmission reliability and data accuracy while considering energy consumption. Recognizing that data accuracy is influenced by the number of intermediate aggregation nodes, our approach aims to minimize this count. To achieve this goal, the measurement methods for transmission reliability and data accuracy are defined first. Taking transmission reliability and path energy consumption as the optimization objectives, an accuracy constraint-based optimization model is formulated to transform the routing problem into a multi-objective optimization problem with the help of a fitness function to assess the solutions and identify the optimal Pareto solution set for the algorithm. To enhance optimization performance, the Improved Adaptive Artificial Immune Algorithm (IAAIA) [16] is employed. To adapt IAAIA to multipath routing, multiple adaptive immune operators are introduced, which not only retain the search capabilities of the artificial immune algorithm but also harness the adaptive nature of immune operators to avoid “premature” convergence. Experimental results show that the proposed routing algorithm can efficiently find a solution to multipath within a reasonable convergence time, meeting the criteria for transmission reliability and data accuracy.

The main contributions of this work include:A multipath routing algorithm EMRAR is proposed. Unlike existing schemes that either construct a congestion-free transmission route to guarantee data accuracy at the expense of reliability or solely ensure data accuracy at the source node, EMRAR can concurrently guarantee transmission reliability and data accuracy at the destination node. It accomplishes this by efficiently controlling the number of paths and aggregation nodes to minimize accuracy loss.A constraint-based multi-objective optimization formula is constructed. This formula considers both transmission reliability and path energy consumption as optimization objectives under the constraints of accuracy loss and node residual energy. This approach ensures a balanced trade-off between these critical factors.The IAAIA algorithm is adopted and customized. To address the multi-objective optimization problem and apply it to the multipath routing scheme, the adaptation includes the design of an antibody initialization method and enhancements to the antibody incentive calculation method and immune operation. These modifications optimize the algorithm’s effectiveness in the context of multipath routing.

The organization of the rest of this paper is arranged as follows. In Section 2, the existing research on multipath routing is discussed, with a particular focus on examining various multipath construction methods. In Section 3, the system model is established to obtain the reliability and energy optimization objectives while adhering to the accuracy loss constraint. In Section 4, the solution for the multi-objective optimization is proposed. The antibody population update strategy is designed, and the fitness function is constructed to evaluate the antibody individuals. In Section 5, the time complexity and message complexity of the algorithm are analyzed. In Section 6, the simulations are carried out to compare EMRAR with existing algorithms. Finally, the conclusions and future work are given in Section 7.

## 2. Related Work

### 2.1. Multipath Routing in WSNs

Multipath routing techniques have found widespread application in enhancing the delivery of multimedia content, providing fault-tolerance routing, and supporting QoS across diverse network environments, including multihop Local Area Networks (LANs), Wireless Area Networks (WANs), the Internet, as well as ad hoc networks and WSNs [17]. The advantages of various multipath routing protocols for WSNs and their associated benefits were presented in [18]. Both best-effort and real-time multipath routing protocols for Wireless Multimedia Sensor Networks (WMSNs) are investigated in [19] from a network application perspective.

The performance of the Multipath Routing Protocol (MRP) is heavily reliant on both the quantity and quality of selected paths. Consequently, MRPs are designed with distinct principles, primarily based on whether the paths have intersection nodes or links, and can generally be categorized into three types: node-disjointed multipath, link-disjointed multipath, and intersection-based multipath. Disjointed multipaths are particularly popular because there is no node or link overlap among paths, which greatly mitigates the impact of node failure or link disconnection on the overall routing functionality. This characteristic offers substantial advantages in prolonging network lifetime and enhancing fault tolerance. Extensive research has been devoted to discovering multiple disjointed paths among multiple nodes over the past two or three decades. Although some algorithms only provide two disjointed paths [20,21,22,23], others can offer more paths [24,25].

Generally, routing protocols used for real-time applications typically impose stringent demands for various QoS metrics such as delay, throughput, and reliability. In [26], by finding two disjointed paths with the shortest transmission distance between source and destination nodes, the number of relay nodes is limited, which improves the reliability and fault tolerance of the network. Addressing the high maintenance costs associated with disjointed multipath routing, Ref. [27] proposes an enhanced AODV protocol. Recognizing that delay-sensitive data, integrity-sensitive data, and ordinary data impose varying QoS requirements on routing, Ref. [28] allocates reserved disjointed multipath to different data types according to the QoS capabilities of different paths, ensuring high delivery rates for integrity-sensitive data and low latency for delay-sensitive data. To mitigate energy consumption and prolong the lifetime, an adaptive multipath routing scheme based on balanced energy consumption is proposed in the literature [29]. This approach jointly considers the path energy, distance, hops, and the lowest energy node, dynamically adjusting the amount of data transmission among multiple paths according to the change in energy information entropy of each path. Meanwhile, in [30], the authors present a multipath routing algorithm to prolong the lifetime by effectively balancing the offered load among forwarding nodes.

However, building completely disjointed multipaths in sensor networks can be challenging in practical scenarios. As demonstrated in [31], the effort to find a multipath routing solution with more than two disjointed paths can incur significant overhead, encompassing computational complexities, energy consumption, and storage demands, especially in sensor networks with only limited resources. Ref. [32] proves that it is NP-hard for a source node to identify more than three disjointed paths. To address this challenge, Ref. [27] optimizes disjointed multipath routing by setting aggregation nodes in networks. The aggregation nodes allow paths to intersect, therefore enhancing routing scalability. Furthermore, Ref. [8] augments the number of paths to improve transmission reliability by constructing braided paths around disjointed multipaths from source to destination. It becomes evident that compared to disjointed multipath routing, intersection-based multipath routing can provide more paths and is more straightforward to implement.

### 2.2. Data Accuracy Guarantee in WSNs

According to the locations within the network where data accuracy guarantee tasks are performed, existing research can be categorized into two distinct schemes: transmission-based and source node-based schemes.

For the transmission-based scheme, as exemplified by literature [15], a congestion control algorithm is introduced to address the problem of substantial data accuracy loss in existing congestion control algorithms. This algorithm operates under the premise that nodes transmit data upward to the sink through a routing tree. Given the upbound error at the destination node, the maximum allowable error for each subtree can be calculated based on the tree’s topological attributes. When congestion occurs, nodes employ lossy compression to compress data within the maximum tolerable error range, which can mitigate the congestion while preserving data accuracy to a certain extent at the destination node. Similarly, in the literature [33], accuracy loss is reduced by integrating a clustering algorithm with a routing algorithm, therefore diminishing the number of data copies in the network. This algorithm leverages a minimum spanning tree-based routing algorithm to transmit data copies to the destination node. In this scheme, cluster heads aggregate copies within their respective clusters to reduce the number of copies. However, the tree-based routing approach in this context is less reliable.

In the source node-based scheme, existing algorithms try to ensure accuracy during the data collection. For instance, in the literature [14], spatial-temporal correlations of sampling data are harnessed to reduce the volume of transmitted data and ensure the data accuracy. This approach then integrates multipath routing with a clustering algorithm to ensure accuracy during transmission. However, it cannot address the issue of accuracy loss caused by data aggregation in multipath transmission. In [34], the authors proposed an AUV-aided hybrid data collection scheme based on the Value of Information (VoI). It tries to keep the VoI of urgent information generated at the source node by assigning it a higher transmission priority. Nevertheless, it attenuates the VoI of normal data, potentially compromising the accuracy of all data.

It can be seen that the transmission-based scheme struggles to deliver highly reliable transmissions to the destination, while the source node-based scheme faces challenges in eliminating the accuracy loss during transmission or may focus solely on specific data types. In a large-scale sensor network employing multipath routing to ensure data transmission reliability and considering the constraints of limited resources, adopting intersection-based multipath routing appears reasonable. Therefore, data aggregation at intersection nodes is inevitable, and consequently, controlling the number of aggregation nodes becomes crucial to reduce the accuracy loss caused by multiple aggregations.

## 3. Multi-Objective Optimization Model for EMRAR

To design the EMRAR algorithm, we formulate the task of identifying a multipath set as a multi-objective optimization subject to constraints, which jointly optimize the objectives of reliability and energy consumption while adhering to the constraint of accuracy loss, which translates to reducing the number of intersection nodes. The construction of the optimization model is detailed as follows.

The coding scheme will not be specified and analyzed as it falls outside the scope of the focus of this work.

For ease of expression, the terms “node”, “sensor”, or “sensor node” will be interchangeably used to refer to a sensor node in the subsequent sections of this paper.

Consider a wireless sensor network system with *n* sensor nodes randomly distributed in the monitoring area. Due to the limitation of the communication distance, all nodes send their collected data to the sink via a multihop mode. To facilitate our discussion, we establish the following definitions based on Graph Theory.

Weighted Undirected Graph *G*: a WSN can be illustrated as a weighted undirected graph *G* = {*V*, *E*}, where *V* = {*v*_1_, *v*_2_, …, *v_n_*} is the set of sensor nodes, and *E* = {*e*_1_, *e*_2_, …, *e_k_*} is the set of edges between two sensor nodes.Neighbor: Suppose the communication range of each sensor node is R. Let the distance between sensor nodes be *i*∈*V* and *j*∈*V* be *d*(*i*, *j*). If *d*(*i*, *j*) ≤ *R*, then nodes *i* and *j* are neighbors, and there is an edge between them. The neighbor relationship can be represented as *i*∈*N*(*j*) or *j*∈*N*(*i*), where *N*(*i*)⊂*V* and *N*(*j*)⊂*V* are the neighbor sets of *i* and *j*, respectively.Multipath Set *MP*(*i*, *S*): Suppose a source node *i*∈*V* can set up multipath to the sink *S*∈*V*. Then, the set of all the paths from *i* to *S* can be represented as *MP*(*i*, *S*). $\left|MP(i,S)\right|$ is the number of paths of *MP*(*i*, *S*). Each path in *MP*(*i*, *S*) can be represented as *mp_j_*(*i*, *S*), *j* = 1, …, |MP(i,S)|. $\left|{mp}_{j}(i,S)\right|$ is the number of nodes in path *mp_j_*(*i*, *S*).

### 3.1. Optimization Objects

Reliability object

Reliability is evaluated by the transmission success rate (TSR) of the data transmitted from a source node to the sink by multiple paths. It is assumed that bit errors and packet losses are the primary factors impacting the transmission success rate. The TSR can be calculated as follows.

For bit error

Suppose the data transmission between neighbor nodes is a single hop, and the bit error rate (BER) in the single-hop wireless channel at node *k* is $e_{k}$. Then, the probability of correctly transmitting one bit to the next hop of node *k* is ${(1-e}_{k})$. Then, the probability of correctly transmitting a packet with *L* bits on one hop at node *k* is
(1)$$p_{k}={\left(1-e_{k}\right)}^{L}$$

For ∀*j*, as *mp_j_*(*i*, *S*) has $\left|{mp}_{j}(i,S)\right|$ hops, then the probability of correctly transmitting a packet from *i* to *S* on *mp_j_*(*i*, *S*) is
(2)$$p_{{BER}_{j}}=\prod_{k=1}^{\left|{mp}_{j}(i,\;S)\right|}p_{k}$$

Then, the average probability of correct transmission on BER is
(3)$$p_{BER}=\frac{1}{\left|MP(i,\;S)\right|}\sum_{j=1}^{\left|MP(i,\;S)\right|}p_{{BER}_{j}}$$

For packet loss

Suppose the packet loss rate (PLR) for a path *mp_j_*(*i*, *S*) is $q_{j}$, then the probability of successfully transmitting a packet from *i* to *S* on *mp_j_*(*i*, *S*) is $\left(1-q_{j}\right)$. Hence, the average probability of successful transmission of PLR is
(4)$$p_{PLR}=\frac{1}{\left|MP(i,\;S)\right|}\sum_{j=1}^{\left|MP(i,\;S)\right|}\left(1-q_{j}\right)$$

For a packet, if it does not arrive at the sink properly, either bit error or packet loss happens. If both happen, it can be treated as packet loss. Then, the average successful transmission probability $p_{h}$ is
(5)$$p_{h}=max\{p_{BER},p_{PLR}\}$$

To meet the demand for reliability, more paths are required. Suppose the paths required to deliver *M* coding packets to the sink is $\left|MP(i,S)\right|$ ($\left|MP(i,S)\right|$≤*M*). For each packet, the successful transmission probability is $p_{h}$ and the failure probability is $\left(1-p_{h}\right)$. The *TSR* for transmitting these *M* packets on $\left|MP(i,S)\right|$ paths is
(6)$$TSR=\sum_{m=1}^{M}C_{\left|MP\left(i,S\right)\right|}^{m}{\left(1-p_{h}\right)}^{\left(\left|MP\left(i,S\right)\right|-m\right)}p_{h}^{m}$$

To optimize *TSR*, we have
Obj1: Max(*TSR*)(7)

2.Energy consumption object

In a sensor network, energy consumption is a critical factor of concern. Having a greater number of paths increases reliability, but it also results in higher energy consumption. Suppose the energy consumption for packet transmission from one node to its neighbor is *E_p_*. Then, the energy consumption for transmitting a packet on path *mp_j_*(*i*, *S*) is
(8)$$E_{j}=E_{p}\sum_{k=0}^{\left|{mp}_{j}\left(i,\;S\right)\right|-1}p_{k}$$

Therefore, on the multipath transmission with |MP(i,S)| paths, the total energy consumption is
(9)$$E_{m}=\sum_{j=1}^{\left|MP\left(i,\;S\right)\right|}E_{j}=E_{p}\sum_{j=1}^{\left|MP(i,\;S)\right|}\sum_{k=0}^{\left|{mp}_{j}\left(i,\;S\right)\right|-1}p_{k}$$

To optimize *E_m_*, we have
Obj2: Min(*E_m_*)(10)

### 3.2. Constraints

Accuracy constraint

Accuracy is used to evaluate the variance between the data collected and sent by the source node and the corresponding data received at the sink. Data aggregation predominantly takes place at path intersections during data transmission. For a given path, as explained in the previous section, the fewer intersections it has with other paths (or less aggregation occurs), the smaller the decrease in accuracy caused by data aggregation during data transmission. To quantify this accuracy decrease, we introduce three metrics: NAL (Node Aggregation Loss), PAL (Path Accuracy Loss), and MAL (Multipath Accuracy Loss), defined as follows:NAL: An NAL *δ*(0 ≤ *δ* ≤ 1) means that the approximation transmission error on an aggregation node on path *mp_j_*(*i*, *S*) is within a factor of *δ*.PAL: Suppose a path *mp_j_*(*i*, *S*) (*j* 1, …, |MP(i,S)|) has $c_{j}^{\left(i\right)}$ intersections with other paths in *MP*(*i*, *S*). If the accuracy loss for each aggregation is *δ*, then for each path *mp_j_*(*i*, *S*), the PAL ${∆}_{j}^{\left(i\right)}$ is:(11)$${∆}_{j}^{\left(i\right)}=1-{\left(1-\delta \right)}^{c_{j}^{\left(i\right)}}$$MAL: Suppose the data collected and sent by sensor node *i* is *x_i_*. After multipath transmission, the data received by sink *S* is *x_i_’*. If the transmission is successful on each path in *MP*(*i*, *S*), then the MAL ${∆}^{\left(i\right)}$ for *MP*(*i*, *S*) from *i* to *S* can be calculated as:(12)$${∆}^{\left(i\right)}=\frac{\left|x_{i}^{\prime }-x_{i}\right|}{x_{i}}=\frac{\sum_{j=1}^{\left|MP\left(i,S\right)\right|}{∆}_{j}^{\left(i\right)}}{\left|MP\left(i,\;S\right)\right|}$$

Hence, for a predefined maximum MAL value Δ, the accuracy constrain can be represented as:(13)$${∆}^{\left(i\right)}=\frac{\sum_{j=1}^{\left|MP(i,S)\right|}{∆}_{j}^{\left(i\right)}}{\left|MP(i,\;S)\right|}\leq \Delta$$

To satisfy Formula (13), the multipath routing algorithm must try to find multiple paths with fewer intersection nodes. Thus, it also gives the constraint on the value of $c_{j}^{\left(i\right)}$.

2.Node energy constraint

As data transmission and aggregation all consume energy, only the nodes with sufficient residual energy can participate in multipath routing. Suppose the residual energy for a node *k* in a path *mp_j_*(*i*, *S*) (*j* = 1, …, |MP(i,S)|) is *E_k_* (*k* = 1, …, $\left|{mp}_{j}(i,S)\right|$). The initial energy of node *k* is *E_max_,* and the predefined minimum residual energy required for a relay node in a path is *E_min_*. Accordingly, we have
(14)$${E_{min}\leq E}_{k}\leq E_{max},k=1,\;...,\;|{mp}_{j}(i,S)|$$

### 3.3. Multi-Objective Optimization Function

To design the multi-objective function, the reliability object Formula (7) needs to be transferred to an equivalent minimization function as
(15)Max(TSR) ≡Min(1/TSR)

Then, the evaluation function can be designed as
(16)$$Obj=\omega_{1}\frac{1}{TSR}+\omega_{2}E_{m}$$
where $\omega_{1}+\omega_{2}=1.$ Hence, the multi-objective optimization problem can be formulated as
(17)$$\left\{\begin{array}{c} Min\left(Obj\right)=Min(\omega_{1}\frac{1}{TSR}+\omega_{2}E_{m})\\ Subject\;to\;\\ \omega_{1}+\omega_{2}=1\\ \frac{\sum_{j=1}^{\left|MP(i,S)\right|}\left[1-{\left(1-\delta \right)}^{c_{j}^{\left(i\right)}}\right]}{\left|MP(i,\;S)\right|}\leq \Delta \\ {E_{min}\leq E}_{k}\leq E_{max},k=1,\;...,\;|{mp}_{j}(i,S)| \end{array}\right.$$

It can be seen that the problem of balancing data accuracy and data transmission reliability has been modeled as a multi-objective optimization. In multi-objective optimization, different objectives exhibit varying value ranges and dimensions, which can potentially lead to conflicts among optimization objectives. For instance, optimizing one objective may inadvertently compromise the performance of another one. Consequently, in the context of multi-objective and multi-constraint optimization problems, the optimization objectives manifest as convex functions with optimal solutions. Generally, there is no universal optimal solution that satisfies all objective functions. Therefore, the introduction of non-inferior solutions also referred to as satisfactory solutions or Pareto optimal solutions, becomes imperative. The outcome of multi-objective optimization is to generate a set of mutually balanced solutions where the target values of these solutions are non-inferior to one another while still meeting the specified constraints.

## 4. Multi-Objective Optimization Resolution

In this section, we tackle the solution to the constrained multi-objective optimization problem (16). In addition to classical algorithms, recent years have witnessed a surge in the popularity of intelligent algorithms, including evolutionary algorithms [16,35]. In this context, an Improved Adaptive Artificial Immune Algorithm (IAAIA) [16] is adopted, which is an enhancement of the classic Artificial Immune Algorithm (AIA) [36]. IAAIA not only inherits the search characteristics of the AIA algorithm but also exhibits robust global search capabilities. Furthermore, it improves convergence speed, accuracy, and stability, all of which are achieved without introducing additional computational complexity to the algorithm.

The IAAIA framework encompasses several key steps, including antigen and antibody construction, initial antibody group formation, affinity calculation, immune operations, antibody group updates, and criteria for algorithm termination. However, IAAIA is primarily geared towards mathematical optimization and cannot be directly applied to the optimization of routing schemes in this work. To apply IAAIA to multi-objective optimization (16), we must customize the antibody initialization method, antibody incentive calculation method, and immune operations while retaining its advantages of fast convergence and precise computation. Detailed calculations and modifications are described in the following subsections.

### 4.1. Antibody Coding

In the immune algorithm, antibody represents the feasible solution to the optimization problem. In the routing scheme, an antibody represents a group of paths from the source node to the destination node, one of which is a gene of the antibody. Hence, suppose the candidate multipath set from source node *i* to sink *S* is *C*, which is generated by the following Algorithm 1.
**Algorithm 1. Candidate multipath set generation**
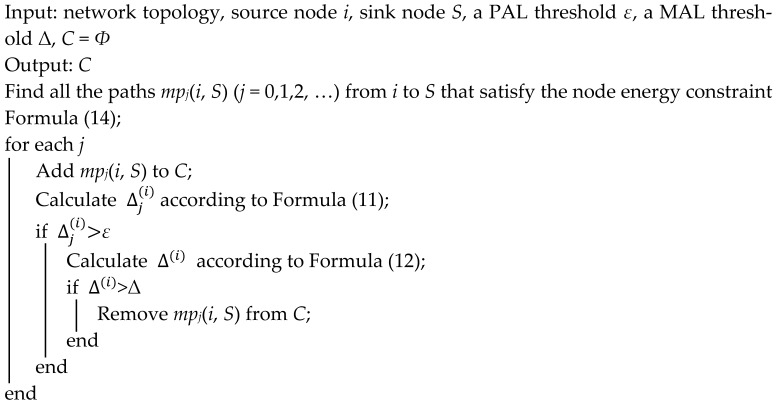


The number of paths in *C* is *L*. Then the antibody has *L* genes. Extract *l* (*l* ≤ *L*) paths from *C* randomly to form a subset *C*′⊆*C*. Let *C*′ be the initial antibody population. Then, for the *L* genes, there are *l* non-zero genes and *L* − *l* zero genes, where each non-zero gene corresponds to a path from the source node *i* to the sink node *S*. Because the hops of each path may be different, variable length coding scheme is adopted in this algorithm. As an example, shown in Figure 1, source node *i* has multipath to the destination *S* via nodes set {1, 2, 3, 4, 5, 6, 7}. Suppose an antibody is composed of three genes. As a path is a gene, the coding of the antibody is {3→4→5, 1→2, 6→7→2}. Based on this coding scheme, the number of joint nodes in this antibody and the hops of each path can be calculated, and then the accuracy loss, energy consumption, and reliability of the antibody can also be calculated.

### 4.2. Fitness Function

In the immune algorithm, an antigen represents the problem to be optimized, and the fitness function serves as a foundational tool for computing the individual fitness values. These fitness values gauge the compatibility or closeness between an antibody and an antigen, a concept akin to the affinity observed in genetic algorithms. As an antibody is composed of *L* genes, the number of non-zero genes in each antibody may be different, and then the corresponding energy loss *E_m_* and transmission success rate *TSR* are different. According to Formula (16), the individual fitness function of the antibody can be expressed as:*FIT* = 1 − *Obj*(18)

Since the smaller the *Obj* value is, the better the optimization effect is, and the larger the fitness *FIT* is, the better the antibody performance is.

### 4.3. Antibody Simulation Calculation

Antibody simulation serves as a comprehensive metric for evaluating the quality of antibodies. It encompasses both fitness and antibody density, with the latter being a crucial measure of antibody diversity within the population. Suppose $f_{aff}(C_{u}^{\prime }{,C}_{v}^{\prime })$ be the affinity between antibodies $C_{u}^{\prime }\subset C^{\prime }$ and $C_{v}^{\prime }\subset C^{\prime }$. As the affinity indicates the similarity between antibodies, the value of $f_{aff}(C_{u}^{\prime }{,C}_{v}^{\prime })$ can be measured by the number of the same paths in these two antibodies. Hence, the same paths set in $C_{u}^{\prime }$ and $C_{v}^{\prime }$ can be expressed as the intersection of two sets and $f_{aff}(C_{u}^{\prime }{,C}_{v}^{\prime })$ can be calculated as:(19)$$f_{aff}\left(C_{u}^{\prime }{,C}_{v}^{\prime }\right)=|C_{u}^{\prime }{\cap C}_{v}^{\prime }|$$

Set *ρ*(0 ≤ *ρ* ≤ 1) as the threshold of the similarity, then judge whether two antibodies are similar or not can be transferred to a 0/1 binary judgment
(20)$$S(C_{u}^{\prime }{,C}_{v}^{\prime })=\left\{\begin{array}{c} 0,\;f_{aff}(C_{u}^{\prime }{,C}_{v}^{\prime })/L\leq \rho \\ 1,\;{\;\;f}_{aff}(C_{u}^{\prime }{,C}_{v}^{\prime })/L>\rho \end{array}\right.$$

The density of antibody $C_{u}^{\prime }$ can be calculated as:(21)$$f_{den}(C_{u}^{\prime })=\frac{1}{N}[1+\sum_{v=1}^{M}S(C_{u}^{\prime }{,C}_{v}^{\prime })]$$
where *N* is the size of the population, and *M* = *N* − 1 is the size of the population after removing antibody $C_{u}^{\prime }$.

Then an antibody $C_{u}^{\prime }$ with higher affinity and lower concentration has higher simulation, which can be calculated as the composition of its fitness and antibody density:(22)$$f_{sim}\left(C_{u}^{\prime }\right)=\beta \frac{FIT\left(C_{u}^{\prime }\right)}{\sum_{u=1}^{N}FIT\left(C_{u}^{\prime }\right)}-(1-\beta )\frac{f_{den}(C_{u}^{\prime })}{\sum_{u=1}^{N}f_{den}(C_{u}^{\prime })}$$
where *β*(0 < *β <* 1) is the weight coefficient of fitness, which is related to the number of iterations and can be calculated as:(23)$$\beta =1-\frac{{(G}_{max}-G_{min})}{G_{max}}\cdot \frac{G}{G_{max}}$$
where *G_max_* and *G_m__in_* are the maximum and minimum number of iterations, respectively. *G* is the current number of iterations. *β* represents the impact of previous-generation antibodies on contemporary antibodies. As the number of iterations increases, the value of *β* linearly decreases, which enables the algorithm to have a high global search ability in the early stage to obtain suitable antibodies while having a high local search ability in the later stage to improve convergence accuracy.

### 4.4. Population Updating

The population update operation comprises three essential steps: antibody cloning, antibody mutation, and antibody updating. This operation continuously refreshes the antibody population until the specified termination condition is satisfied.

Antibody cloning

To speed up the convergence of the population, the antibody with a high incentive should be selected for cloning. The key step involves calculating the average antibody incentive level across the population and subsequently selecting antibodies with incentives exceeding this average for cloning.

2.Antibody mutation

Mutation operation means that in the cloned antibody population, the antibody mutates according to the probability *p_m_* of producing a new antibody. This operation can improve the local search performance of the algorithm and find potential antibodies. The variation probability *p_m_* can be expressed as:(24)$$p_{m}=\left\{\begin{array}{c} \frac{k_{1}(f_{max}-f_{d})}{f_{max}-f_{avg}},\;f_{d}\geq f_{avg}\\ k_{2},\;f_{d}<f_{avg} \end{array}\right.$$
where *f_max_* and *f_avg_* are the maximum value and average value of the antibody fitness, respectively. *f_d_* is the fitness of the mutated antibody. *k*_1_ and *k*_2_ are constants.

During this process, mutation introduces spontaneous random changes in antibodies. Its purpose is to maintain population diversity and prevent the optimization from falling into local optima. The process follows the coding rules of the algorithm as outlined below.

Select an antibody from the antibody population with the mutation probability *p_m_*. Select one of the genes and select a node on this gene as the mutation node.The path from the source node *i* to the mutation node remains unchanged, and the mutation node searches the path to the sink node *S* again to find a new path.Get new antibodies.

3Antibody updating

The population updating operation is to eliminate the antibodies with a low incentive degree in the antibody population and retain the first *N* antibodies with a high incentive degree to form a new population through the above operation.

The framework of the multi-objective optimization solution is illustrated in the flowchart in Figure 2.

### 4.5. EMRAR Algorithm Procedure

The overall procedure of EMRAR is described by the following Algorithm 2. It illustrates the steps of establishing a multipath between a sensing node *i* and the sink node *S*. The |MP(i,S)| routes discovery can be realized by source routing protocols like DSR [37], in which request and reply messages will be sent among nodes between *i* and Shop by hop. Then, the values of $p_{{BER}_{j}}$ and $q_{j}$ can be obtained by node *i* from the reply messages, therefore calculating the values of $p_{BER}$ and $p_{PLR}$.

The EMRAR algorithm operates round and round, similar to LEACH [38], with the goal of achieving a balanced energy consumption among nodes. However, there is a notable distinction between the two approaches. In LEACH, each cluster head enters a sleep state automatically in the next round, whereas in EMRAR, the states of nodes on the multipath in the current round do not enter a sleep state automatically in the subsequent round. Instead, the state is determined based on their residual energy during the initialization process in the subsequent round. The duration for each round can be predefined, for example, 15 min, or adjusted to align with the realistic network environment.
**Algorithm 2. Find optimal multipath by IAAIA**Step 1Network initialization: Each node exchanges residual energy, location, and other information with its neighbors to establish a neighbor table and determine relevant parameters.Step 2Path discovery: when sensing node *i* needs to transmit data to sink node *S*, it sends path probing messages to neighboring nodes. After *S* receives all probing messages, the paths are initialized as a candidate path set if they satisfy the node energy constraint.Step 3Encoding: *S* filters the paths by accuracy constraint; establishes antigen and antibody; encodes them to generate the initial antibody population.Step 4Path selection: calculate the fitness of each path; select paths by elitist selection by calculating the simulations of them to form the antibody population.Step 5Path update: update the paths within the population according to the population update rules.Step 6Repeat: repeat Steps 4 and 5 until the algorithm completion condition is met to obtain the optimal path information.Step 7Multipath establishment: *S* returns a response message along the optimal paths. Upon receiving the response message by node *i*, the multipath is established successfully.Step 8Nodes that are not on the established multipath enter a sleep state till the next round of EMRAR.

## 5. Complexity Analysis

### 5.1. Message Complexity

Suppose each node has *σ* neighbors. In the network initialization phase, each node exchanges information with its neighboring nodes once. Therefore, a total of 2*σ* messages are required for a node to complete the initialization. Then, the message complexity is *O*(*σ*). In the path discovery phase, a node also needs to send *σ* messages to its neighbors, and the messages will be forwarded to maximum *ζ* hops. Then, the message complexity is *O*(*σζ*). After the destination node obtains the optimal paths, it will send back a response along each path in the multipath set. Then, the message complexity is *O*(1). Based on the above analysis, the message complexity is *O*(*σ* + *σζ*) = *O*(*σζ*).

### 5.2. Time Complexity

Suppose each node has *σ* neighbors, and at most *σ* paths will be initialized. The time complexity of this step is *O*(*σ*). The time complexity of the IAAIA algorithm is determined by the immune operation and the number of iterations. The time complexity of crossover in each generation is *O*(*σ*^2^). The fitness values of all paths need to be calculated. Thus, the time complexity of mutation in each generation is also *O*(*σ*). If the number of iterations is *η*, then the total time complexity of EMRAR is *O*(*σ*) + *O*(*η* × *σ*) + *O*(*ησ*^2^) = *O*(*ησ*^2^).

## 6. Evaluation

### 6.1. Simulation Configuration

In this simulation, the configuration of the main parameters is listed in Table 1.

The sensor energy consumption adheres to the classic model described in [38] (the sensors are powered by batteries and do not benefit from alternative energy supply, such as wireless-powered communication technology [39,40]). The energy consumption by a node to transmit *k* bits of data over l meter is:(25)$$C_{s}\left(k,l\right)=E_{elec}k+\varepsilon_{amp}kl^{2}$$
where the radio dissipates $E_{elec}=50\;nJ/bit$ to run the transmitter or receiver circuity and $\varepsilon_{amp}=0.015\;nJ/(bit\cdot m^{2})$ is the transmit amplifier.

For a receiving node, the energy consumption to receive *k* bits of data is:(26)$$C_{r}\left(k\right)=E_{elec}k$$

Hence, on the link *e* between two adjacent nodes, the energy consumption *C*(*e*) is the sum of the energy consumption for transmitting *C_s_* and the energy consumption for receiving *C_r_*, which can be expressed as:(27)$$C\left(e\right)=C_{s}\left(k,l\right)+C_{r}\left(k\right)$$

Based on the energy consumption model, the following performance indexes are used to evaluate the performance of the EMRAR algorithm:Average Accuracy Loss (AAL): the relative error between data sent by the source node and the data received and recovered by the destination node. The AAL value in this work is the average loss of *m* data received during the simulation time, which can be calculated as:
(28)$$AAL=\left(\frac{1}{m}\sum_{i=1}^{m}\frac{x_{i}-x_{i}^{\prime }}{x_{i}}\right)\times 100\%$$
where *x_i_* is the data sent out by the source node, and $x_{i}^{\prime }$ is the data received and recovered by the receiving node.

Packet Delivery Rate (PDR): the ratio of the number of packets received by the destination node *Num_R_* to the number of packets sent by the source node *Num_S_*, which can be calculated as:


(29)
$$PDR=\frac{{Num}_{R}}{{Num}_{S}}\times 100\%$$


Network Residual Energy (NRE): the ratio of the residual energy of all the nodes in the network to the initial energy of the network, which can be calculated as:


(30)
$$NRE=\frac{\sum E_{re}}{\sum E_{ini}}\times 100\%$$


The above indexes are evaluated under different densities of adjacent nodes (average number of neighbors of a node) or different data transmission rates. The experiments are performed in OMNet++ by means of the MiXiM framework. It uses the topology generator to generate random topologies for simulation each time. The experiments are repeated till the results can be achieved at a precision of 1% with a 90% confidence interval. The maximum number of iterations is set to 100.

### 6.2. The Effect of Accuracy Constraints on the Performance of the Algorithm

In this section, the impact of the accuracy constraint ∆ on the EMRAR algorithm is analyzed under different values of ∆ when the data rate is 50 kbps. In line with prior research [11,41], we set the accuracy constraint ∆ with 3 distinct values—1%, 3%, and 5%—in each simulation, representing incremental levels of allowable accuracy loss. The AAL, PDR, and NRE performances in different node densities are shown in Figure 3.

Figure 3a shows the AAL performance under different densities of adjacent nodes. From the figure, we can find 3 patterns:A larger value of ∆ corresponds to a larger value of AAL. This correlation can be attributed to the fact that, with a fixed network size, a larger accuracy constraint ∆ implies a reduced constraint on the paths available for multipath routing. Consequently, the probability of path intersections increases, leading to more data aggregation and ultimately resulting in a higher AAL.An increasing density of adjacent nodes leads to a lower AAL, signifying improved data accuracy. The reason for this is that a higher number of neighbors provides more available next-hop nodes for routing. The algorithm can choose the routes that lead to less aggregation towards the sink, consequently reducing accuracy loss.The reduction in AAL becomes more obvious with a larger value of ∆. The reason for this is that when there is a greater number of neighbors, a larger value of ∆ provides a relatively greater number of alternative nodes. Hence, it becomes easier to find better path choices that significantly reduce the accuracy loss.

Figure 3b shows the PDR performance under different densities of adjacent nodes. The shapes of all three curves are quite similar. For example, when ∆ = 3%, the PDR increases by approximately 4.5% for every 5-unit increase in the number of adjacent nodes. This trend occurs because as the density of adjacent nodes increases, there are more available paths that meet the constraints. This results in a greater number of reliable paths for data transmission, fewer intersection nodes, and reduced congestion in the MAC layer. As a result, the PDR is improved. On the other hand, when the density of adjacent nodes is held constant, increasing ∆ relaxes the constraint on the path and increases the number of available paths, also resulting in improved PDR.

Figure 3c shows the NRE performance under different densities of adjacent nodes after 500 rounds. It can be seen that:Across various values of density, the energy consumption at ∆ = 5% consistently demonstrates superior performance compared to other values of ∆. The reason for this is that when the accuracy constraint is relaxed, more intersection nodes are permitted in the multipath, allowing the establishment of shorter routes (with fewer hops) for data transmission, which can save the energy consumption of the whole network and prolong the lifetime.With the increase in node density, the NRE exhibits a consistent increase across various values of ∆. The reason for this is that higher density implies a larger number of nodes within the area, and the percentage of active nodes relative to the total number of nodes decreases, resulting in an elevated NRE ratio.

The AAL, PDR, and NRE performances in different data transmission rates are shown in Figure 4, with an adjacent node density of 20.

Figure 4a shows the AAL performance under different data transmission rates. It can be seen that as the data transmission rate increases from 50 kbps to 100 kbps, the AAL increases consistently across all ∆ values, signifying a reduction in data accuracy. This is because a higher data transmission rate necessitates the delivery of a greater number of data packets. Consequently, additional paths need to be established to facilitate their transmission. Hence, the path intersection will increase, and the data aggregation at intersection nodes will affect the accuracy of the data received at the sink. Furthermore, at the same data transmission rate, increasing the accuracy constraint ∆ results in a greater number of established paths, further elevating the likelihood of path intersections and subsequent data aggregation. This, in turn, leads to an increase in data accuracy loss.

Figure 4b shows the PDR performance under different data transmission rates. As we can see, the higher data transmission rate corresponds to lower PDR values, and ∆ = 1% exhibits the lowest PDR compared to other values of ∆. This is because higher transmission rates introduce a higher offered load on the network, leading to increased congestion and a subsequent reduction in PDR. Conversely, when the data transmission rate remains constant, PDR increases with the increasing ∆ values. This is because of the relaxation of accuracy constraints on the path, resulting in many available paths and subsequently improving the performance of data delivery.

Figure 4c shows the NRE performance under different data transmission rates. It can be seen that a higher transmission rate brings a larger volume of data being transmitted that will consume more energy by the sensor nodes, resulting in lower NRE values. Additionally, smaller values of ∆ also correspond to lower NRE values. The reason for this is that smaller values of ∆ restrict the utilization of intersection nodes within the multipath, potentially requiring the transmission paths with more hops. Therefore, a greater number of nodes must participate in data transmission to consume more energy and consequently lower NRE values.

### 6.3. Comparison with Other Algorithms

In this section, we conduct a comparative analysis of the EMRAR algorithm against three other algorithms: CADC proposed in [15], AMRBEC proposed in [29], and IQMRP proposed in [35]. Figure 5 presents a performance comparison of these four algorithms in terms of AAL, PDR, and NRE under different data transmission rates with a fixed ∆ value of 3% and an adjacent node density of 20. The NRE is calculated after running each algorithm for 500 rounds.

From these three figures, we can see that as the data transmission rate increases from 50 bps to 100 bps, all four algorithms exhibit a similar trend on AAL, PDR, and NRE. However, for different densities of adjacent nodes, EMRAR consistently outperforms the other three algorithms in most scenarios. The rationale behind this can be explained as follows.

For the AAL in Figure 5a, CADC establishes a tree-based topology rooted at the sink node. It gives upper bounds for estimation error and estimation distortion, which are used to evaluate the accuracy of the data. Although it employs lossy compression to mitigate congestion, it primarily focuses on meeting these upper bounds, potentially resulting in a relatively high AAL if the bounds are not stringent. In contrast, IQMRP adopts an Ant Colony Optimization (ACO) technique to choose the multipath, which, while reducing the number of intersection nodes to some extent, has limited effectiveness in minimizing AAL. For AMRBEC, it consistently establishes a disjointed multipath, and the data are only aggregated at the sink of the node, leading to the lowest data accuracy loss among the four algorithms.

For the PDR in Figure 5b, as the data transmission rate increases, the PDRs of all four algorithms decline because of heightened packet congestion. Among them, CADC does not adopt multipath routing. The single transmission route results in a lower transmission reliability compared to the others. IQMRP chooses multipath based on QoS parameters such as residual energy, bandwidth, and next-hop accessibility. It improves the PDR but with a predefined threshold for path quality evaluation, which is less adaptive. The metrics for AMRBEC to choose the multipath are similar to IQMRP. It is based on the residual energy, distance, hops, and the balance of energy consumption, which are less related to channel quality, leading to more pronounced errors or packet losses than those of IQMRP. EMRAR treats reliability as an optimization objective directly, which can improve the PDR effectively.

For the NRE in Figure 5c, CADC exhibits poor NRE performance due to two primary reasons. First, the routing algorithm is not power-aware, and second, it uses a tree-based topology to transmit data to the sink, which is known to have a disadvantage in balancing energy consumption at the nodes near the sink. IQMRP chooses the multipath by balancing the residual energy, bandwidth, and next-hop accessibility, with residual energy just one of the considerations. AMRBEC initially has the highest residual energy at a lower data transmission rate (<50 kbps), but it degrades as the data transmission rates increase. There are two reasons. First, it must establish the node-disjointed multipath, which involves selecting more nodes, and it is difficult to find shorter paths while adhering to the energy constraints. Second, AMRBEC establishes the multipath without considering the quality of the paths, such as error and packet loss. Thus, the retransmission will consume more energy when dealing with higher offered loads. EMRAR considers both node residual energy and path energy consumption while adhering to the transmission reliability constraint. As a result, it has the best residual energy performance, especially in scenarios with higher offered loads.

Figure 6 provides the comparison of the convergence times under different densities of adjacent nodes. The convergence time is evaluated by the average time required for the network to establish a set of multiple paths from the source to the destination in IQMRP, AMRBEC, and EMRAR or a node to find a route to the sink on a tree-based topology in CADC. It can be seen that as the network size increases, the convergence time of each algorithm increases. However, CADC shows linear growth with increasing adjacent node density, while IQMRP, AMRBEC, and EMRAR show exponential increases. This difference arises because the CADC algorithm only needs to find the shortest path from the source to the destination. It has a linear time complexity to the number of edges in the network. In contrast, IQMRP and EMRAR share similar time complexities—IQMRP adopts the classic ACO algorithm, while EMRAR adopts the IAAIA algorithm that improves the antibody excitation operator of the AIA algorithm and accelerates the convergence. Therefore, EMRAR boasts a shorter convergence time IQMRP. For AMRBEC, which shares the same time complexity as EMRAR, the convergence time curve closely resembles EMRAR’s when the density of the adjacent nodes is larger than 10. However, at lower node densities, AMRBEC may struggle to find a multipath solution with disjointed nodes, resulting in considerably longer average converge times in the simulation. Therefore, AMRBEC is not suitable for scenarios with low node density.

Figure 7 shows the comparison of average end-to-end delay under different data transmission rates. End-to-end delay is evaluated by the average time taken for data to travel from the source to the destination. It can be seen that as the data transmission rate increases, resulting in a higher offered load in the network, the average end-to-end delay of each algorithm increases. Among them, both CADC and AMRBEC algorithms exhibit poorer end-to-end delay performance. For CADC, although it establishes a tree-based topology with the shortest path, data aggregation occurs at each branch node of the tree, leading to increased end-to-end delay caused by channel sharing at branch nodes. On the other hand, AMRBEC, which must establish node-disjointed multipath, faces challenges in establishing multiple shortest paths, occasionally resorting to longer paths. Hence, at lower data transmission rates, AMRBEC fails to outperform CADC in terms of end-to-end delay. However, as the data transmission rate increases due to the node-disjointed multipath, AMRBEC performs better than CADC because of less data aggregation and queuing. IQMRP has the best performance in this comparison because it establishes a QoS-based multipath routing, with latency being one of the measurements. For EMRAR, it prefers paths with small accuracy loss to low-latency next-hop or shortest paths, resulting in an increased end-to-end delay in comparison with IQMRP.

The simulation results reveal distinctive strengths and weaknesses among the evaluated algorithms. CADC is superior in convergence time but poor in all other performance metrics. IQMRP shows the best performance in average end-to-end delay, but it has the largest convergence time and is relatively compromised in other aspects. AMRBEC has minimal accuracy loss and the highest residual energy when the data transmission rate is low. However, its performance is less satisfactory in other aspects, especially when the data transmission rate and node density are low. In contrast, our proposed algorithm, EMRAR, has the best performance in terms of PDR and residual energy while also maintaining competitiveness in other aspects.

## 7. Conclusions

This paper proposed the multipath routing algorithm EMRAR, which aims to strike a balance between transmission reliability and data accuracy in wireless sensor networks. It employs a multi-objective optimization with constraints, utilizing an improved immune genetic algorithm, IAAIA, to obtain the optimal solution. Simulation results verify that the proposed EMRAR algorithm outperforms existing algorithms in terms of transmission reliability, data accuracy, and energy consumption. Convergence time and end-to-end delay are also discussed. The findings suggest that the EMRAR algorithm is well-suited for scenarios with high requirements in data accuracy and transmission reliability at the receiving end, where real-time computing or delay sensitivity is less critical. In addition, the work in this paper provides a valuable method to optimize multiple objectives simultaneously in multipath routing for sensor networks, with potential applications in addressing similar challenges in WSNs.

Our future research in this area will focus on applying the EMRAR algorithm to mobile ad hoc networks, such as VANET, where network topology dynamically changes due to vehicle movements. In such dynamic environments, calculating multiple path sets in real time to satisfy the evolving requirement is crucial. Hence, the impacts of node mobility and the real-time performance of the algorithm are issues that must be further considered.

## Figures and Tables

**Figure 1 sensors-24-00285-f001:**
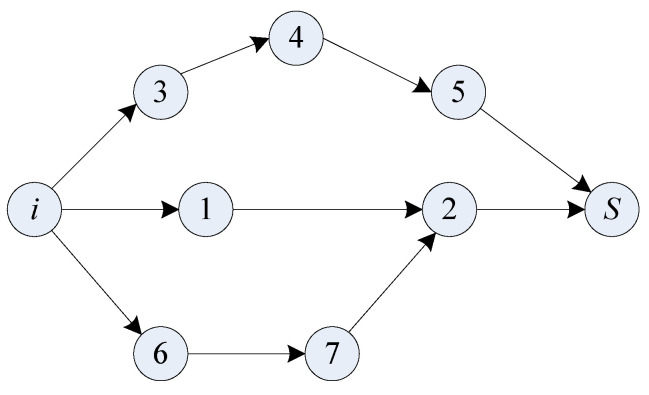
An example of an antibody coding scheme.

**Figure 2 sensors-24-00285-f002:**
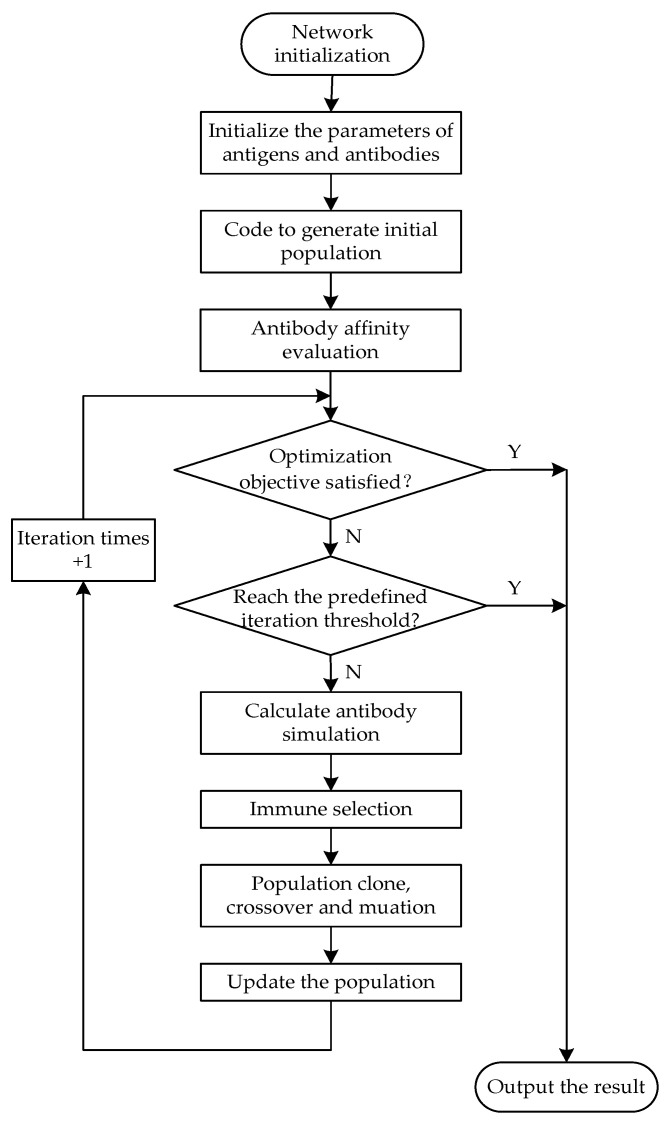
The flowchart for solving multi-objective optimization function.

**Figure 3 sensors-24-00285-f003:**
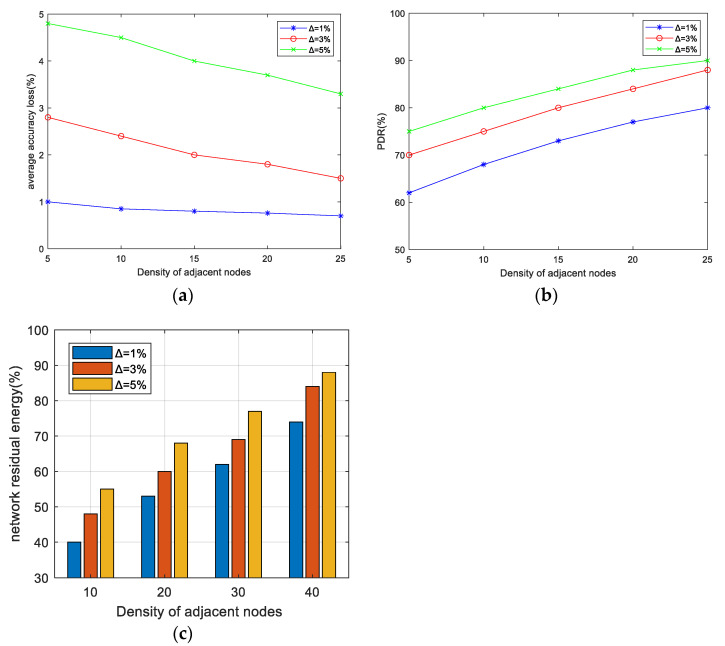
The effect of Δ on AAL, PDR, and NRE under different adjacent node densities when the data transmission rate is 50 kbps. (**a**) AAL performance, (**b**) PDR performance, (**c**) NRE performance.

**Figure 4 sensors-24-00285-f004:**
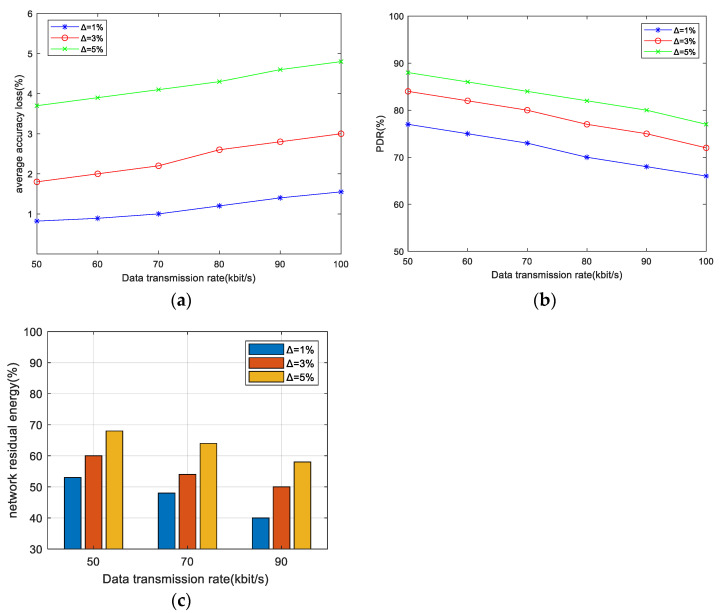
The effect of Δ on AAL, PDR, and NRE under different data transmission rates when the adjacent node density is 20. (**a**) AAL performance, (**b**) PDR performance, (**c**) NRE performance.

**Figure 5 sensors-24-00285-f005:**
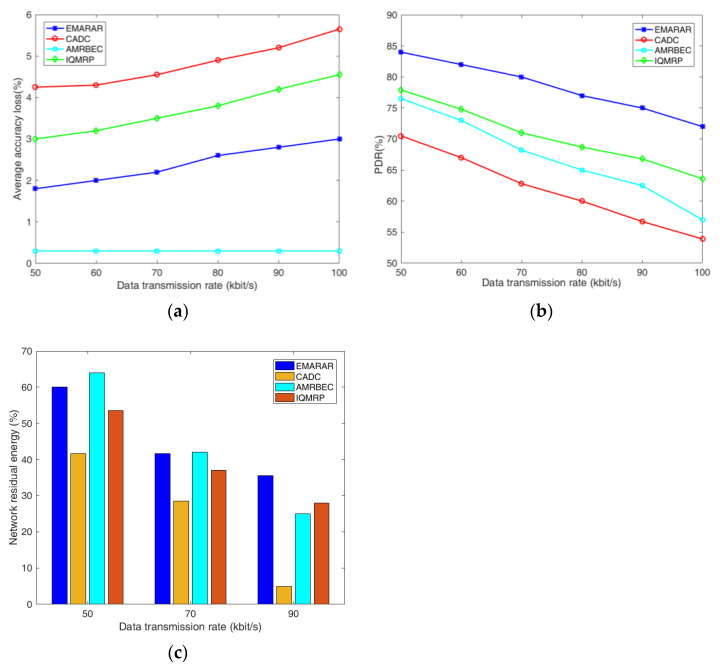
The comparison of three algorithms under different data transmission rates when ∆ = 3% and the adjacent node density is 20. (**a**) AAL performance, (**b**) PDR performance, (**c**) NRE performance.

**Figure 6 sensors-24-00285-f006:**
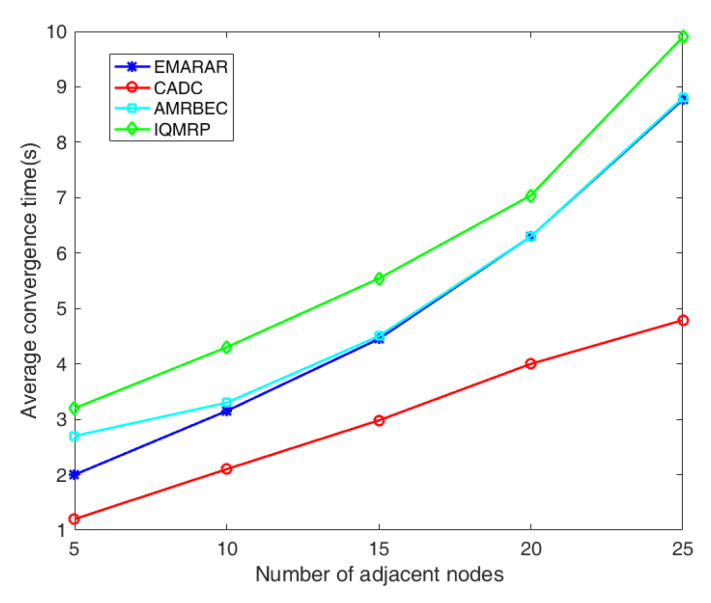
The comparison of three algorithms on convergence time under different adjacent node densities.

**Figure 7 sensors-24-00285-f007:**
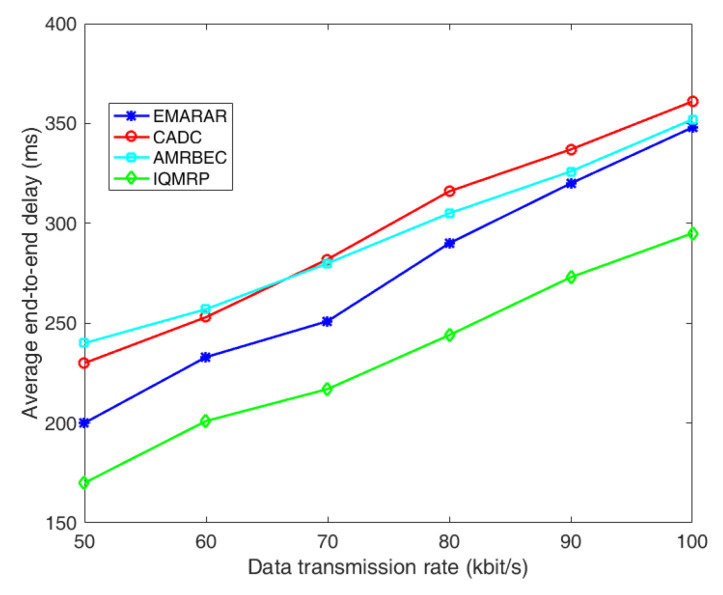
The comparison of average end-to-end delay under different data transmission rates.

**Table 1 sensors-24-00285-t001:** The configuration of main parameters.

Parameters	Configuration
CPU	Core I5 processor
Main frequency	2.5 GHz
Sensing area range	500 m × 500 m
maximum communication range	50 m
Initial energy of sensors	2 J
Bit error rate *e*	random(0, 1] × 10^−6^
Packet loss rate	random(0, 1] × 10^−2^
Node Accuracy Loss *δ*	0.005
MAC protocol	802.15.4
Packet length	16 kbits

## Data Availability

Data are contained within the article.

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
