# Peer review of "An Accuracy-Aware Energy-Efficient Multipath Routing Algorithm for WSNs"

_sensors, 2024, doi:10.3390/s24010285_

Round 1
Reviewer 1 Report
Comments and Suggestions for Authors
In this article, an power-efficient multipath algorithm EMRAR is proposed to balance the reliability and accuracy of data transmission. After my reviewing, the work is well-written and interesting, but it can not be accepted in its current version and some following comments are required to improve the quality of the article.
1. In the abstract, what does “EMRAR” mean?
2. The notation section of this article should be added to make the presentation easy to follow.
3. The contributions of this work are not clear. More explanations or comparisons are needed to show the advantages or differences of the proposed strategy with some existing methods.
4. The literature reviewing is insufficient. Time delay is an important issue for WSNs. Some recent results about this topic are required to be discussed to update the reviewing. For example, A delay-kernel-dependent approach to saturated control of linear systems with mixed delays.
5. There exist some grammar errors and typos in this paper. Please check the paper carefully. For example, in line 56: To solve this problems,…
Comments on the Quality of English LanguageModerate editing of English language required.
Reviewer 2 Report
Comments and Suggestions for Authors
In this work a power-efficient multipath algorithm EMRAR is proposed to balance the reliability and accuracy of data transmission. I have following concerns;
The keywords are not arranges in alphabetical order.
After the line 348, please fix the issue of the table text overlapping.
What is the difference between the work “Situation-Aware Dynamic Service Coordination in an IoT Environment” and your work? Please elaborate some differences.
You can add the summery of the work “On the Ergodic Secrecy Capacity of Intelligent Reflecting Surface Aided Wireless Powered Communication Systems” to more elaborate the wireless power communication.
Why did you select EMRAR algorithm as compared to other existing techniques? Please justify it.
For the validation of your work, please clearly describe the comparison with the existing work.
Reviewer 3 Report
Comments and Suggestions for Authors
The manuscript presents an algorithm to optimize multipath routing in Wireless Sensor networks. It is well-written, and the presentation is good. However, I suggest inserting a table with all variables in the text.
The only issue is about the experiments. I could not find or understand what and how many network topologies were considered. Moreover, the graphs present only the average value. I missed the confidence interval to improve the algorithm evaluation.
Comments on the Quality of English Language
The language is good. I find only a small number of phrases a little bit hard to understand, such as line 166. The overall manuscript is good.
Reviewer 4 Report
Comments and Suggestions for Authors
This article proposes an algorithm - EMRAR and conducts numerical simulations. The authors also compared their simulations with other three algorithms - CADC, AMRBEC, and IQMRD under different data tranmission rates. it looks the article providing a study on multiple routing algorithm for WSNs with solid experimental results.
The English writing of this article exist many grammer errors that must be corrected. i list some of errors below.
1. line 13, the transmision reliability, the data accuracy. no need 'the'
2.line19, constrain, should be 'constraint'
3.line 31, the fields, no need 'the'
4.line40, the transmission reliability. no need the
5.line45, has, should be replaced by have
6.line 56, this problems, should be this problem
7.line 63 source, needs to add the
8.line 133, needs to add the, in the past 2 or3
9.line 166, was executed, or is executed, missing 'was/is'
10.line224, no need 'the', in the all....
11.line 235, missing 'a', in single hop
12.line344, missing the
13.line543, density should be replaced by densities.
..........................................................
There are many such errors across this artcile that the author must correct.
Comments on the Quality of English Languageas above. need to improve English language significantly
